# StreamUni: Achieving Streaming Speech Translation with a Unified Large Speech-Language Model

## Abstract

Streaming speech translation (StreamST) requires determining appropriate timing, known as policy, to generate translations while continuously receiving source speech inputs, balancing low latency with high translation quality. However, existing StreamST methods typically operate on sentence-level speech segments, referred to as simultaneous speech translation (SimulST). In practice, they require collaboration with upstream segmentation models to accomplish StreamST, where the truncated speech segments constrain SimulST models to make policy decisions and generate translations based on pre-defined contextual information preset by the upstream models. Moreover, SimulST models struggle to learn effective policies due to the complexity of speech inputs and cross-lingual generation. To address these challenges, we propose StreamUni, which achieves StreamST through a unified Large Speech-Language Model (LSLM). Specifically, StreamUni incorporates speech Chain-of-Thought (CoT) in guiding the LSLM to generate multi-stage outputs. Leveraging these multi-stage outputs, StreamUni simultaneously accomplishes speech segmentation, policy decision, and translation generation, completing StreamST without requiring massive policy-specific training. Additionally, we propose a streaming CoT training method that enhances low-latency policy decisions and generation capabilities using limited CoT data. Experiments demonstrate that our approach achieves state-of-the-art performance on both SimulST and StreamST tasks.

## 1 Introduction

Streaming speech translation (StreamST) (Ma et al., 2019; 2020b; Dong et al., 2022), known as simultaneous interpretation, generates corresponding translations while continuously receiving incoming source speech inputs. Given its real-time nature, StreamST is commonly employed in various cross-lingual communication scenarios such as international conferences and real-time subtitles.

Compared to traditional offline speech translation (Gangi et al., 2019; Alinejad & Sarkar, 2020; Lee et al., 2022), StreamST must not only ensure translation quality but also minimize the latency between receiving speech inputs and generating translations (Zhang et al., 2024a). To this end, StreamST requires a generation policy to determine the appropriate timing for outputting each translated word. Additionally, considering that StreamST is often deployed in scenarios lasting tens of minutes to several hours (Ma et al., 2019), and that the relevant content attended to by StreamST is primarily concentrated around real-time inputs (Papi et al., 2024), it becomes necessary to implement a truncation policy that can truncate historical speech inputs and translations. This enables the model to focus on recent speech inputs while preventing information overload that could compromise efficiency. Therefore, an ideal StreamST model requires both an effective generation policy and truncation policy to achieve low latency and high translation quality.

Existing methods primarily belong to simultaneous speech translation (SimulST) rather than StreamST, as they cannot be directly applied to speech streams lasting tens of minutes, but are instead limited to speech clips of less than 20 seconds (Tang et al., 2023), which are segmented by upstream modules such as Voice Activity Detection (VAD) (Team, 2024). Due to the short duration of speech clips, current SimulST methods focus on the generation policy, which can be broadly cate-

gorized into fixed policy and adaptive policy. Fixed policy (Ma et al., 2019; 2020b) guides the model to alternately read fixed-duration speech chunks and output a predetermined number of words. This approach, which disregards the actual textual content within the speech, typically leads to redundant latency or poor translation quality. Moreover, adaptive policy employs integrate-and-fire (Dong et al., 2022), CTC (Zhang et al., 2024a), and Transducer (Tang et al., 2023) to determine generation policy based on the text density of the input speech, achieving better performance. However, these methods still deliver suboptimal translation quality due to small-scale Transformer (Vaswani et al., 2017) architectures.

More recent work attempts to leverage the powerful generation capabilities of Large Speech-Language Models (LSLMs) for SimulST, delivering superior performance. These methods either adopt fixed policy (Agostinelli et al., 2024) or adaptive policy achieved by fine-tuning LSLMs with extensively constructed policy-specific data to enable autoregressive policy prediction (Wang et al., 2024; Cheng et al., 2024; Labiausse et al., 2025). However, such fine-tuning methods not only compromise the inherent generation capabilities of LSLMs but also present difficulties in efficiently transferring to newly advanced LSLMs. Therefore, existing SimulST methods face substantial challenges in enabling LSLMs to conduct effective generation policy learning. Furthermore, current research has inadequately explored truncation policies, with attempts to timely truncate historical translations through constructing complex translation trajectory training data and sliding window schemes (Ouyang et al., 2025). This approach not only incurs substantial data construction costs but also hinders seamless transfer to cutting-edge LSLMs. Consequently, investigating the use of a unified LSLM to efficiently implement StreamST has emerged as a highly promising paradigm.

Despite its advantages, implementing StreamST using a unified LSLM remains challenging, as it requires LSLM to simultaneously handle truncation and generation policies while achieving real-time translation. To determine generation policy, LSLMs need to detect valid content in real-time speech stream and decide on the optimal generation timing and output translations (Dong et al., 2022). As the speech stream grows, LSLMs require the truncation policy to discard historical speech segments and translations, ensuring the model focuses on recent inputs while avoiding excessive computational overhead (Papi et al., 2024). Truncation policy must ensure that discarded speech segment is fully translated and that discarded translations accurately correspond to the discarded speech segments, thereby maintaining truncation integrity. Beyond policy decisions, StreamST also needs to accomplish high-quality translation for continuously incoming speech input streams. However, conventional approaches that separately optimize these three subtasks require constructing substantial amounts of corresponding training data (Wang et al., 2024), which is not only resource-intensive but also present significant difficulties in transferring to newly advanced LSLMs. Therefore, investigating how to enable LSLMs to efficiently accomplish all subtasks in a unified manner for effective StreamST is of paramount importance.

To address these challenges, we propose StreamUni, a framework that efficiently enables a unified LSLM to accomplish all subtasks of StreamST in a cohesive manner. StreamUni introduces the speech Chain-of-Thought (CoT) (Huang et al., 2023; Nguyen et al., 2024) that guides LSLMs to progressively generate transcriptions and translations based on the speech inputs. Leveraging multi-stage outputs, the model handles generation policy, truncation policy, and streaming translation generation subtasks. For the generation policy, StreamUni detects effective speech chunks in real-time through intermediate transcriptions to determine optimal generation timing, and decides the current output translation based on the coherence between real-time transcription and previously output translations. For truncation policy, StreamUni maintains transcription queues across different timestamps and determines speech truncation timing by comparing current and historical transcriptions. Once the source truncation point is identified, StreamUni prompts the LSLM to output complete translations for speech segments preceding the truncation point, subsequently discarding the corresponding translations and speech segments to maintain truncation integrity. The real-time translation generation is obtained by selecting appropriate output translation from the speech CoT based on the generation policy. Through this design, StreamUni achieves StreamST via multi-task results across multiple stages of the speech CoT.

To further enhance streaming performance, we propose a Streaming CoT training scheme that optimizes multi-stage CoT outputs by encouraging LSLMs to predict corresponding transcriptions and complete translations based on partial speech inputs. Therefore, StreamUni unifies all subtasks through the speech CoT and achieves holistic optimization via a unified training strategy. Experi-

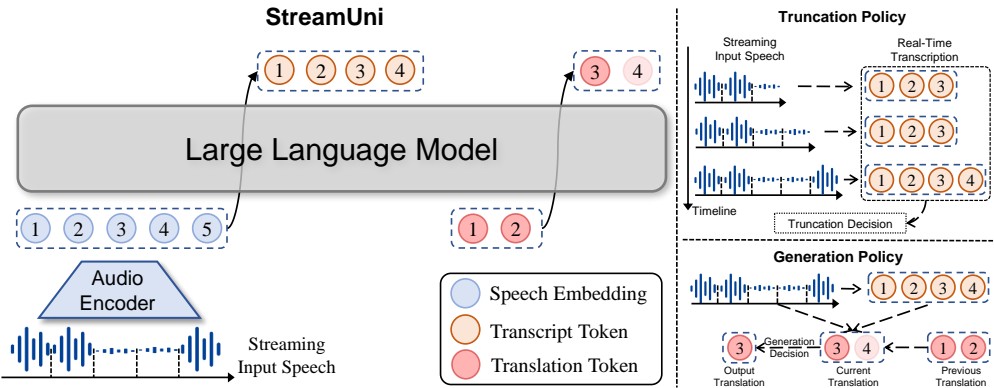

Figure 1: The framework of StreamUni and illustration of truncation policy and generation policy. The model will generate real-time transcription based on the existing speech inputs and compare it with historical transcriptions to determine the truncation policy. If truncation is decided, the model will bypass the generation policy and continue generating the full translation of the current speech. Otherwise, the model will determine the number of translation words to continue generating based on the lag relationship between the real-time transcription and the output translation, and generate the translation using CoT.

ments demonstrate that our method efficiently achieves state-of-the-art performance on StreamST tasks across multiple directions.

## 2 BACKGROUND

**Streaming Speech Translation** Let the complete speech stream be represented as $\mathbf{s} = (s_1, ..., s_N)$, where $s_i$ denotes a speech chunk of predefined size, typically around 320ms or 640ms. Given the continuously arriving input speech chunks, the StreamST model progressively generates translation $\mathbf{y} = (y_1, ..., y_I)$ under a generation policy $\mathbf{g} = (g_1, ..., g_I)$ where $g_i$ represents the number of speech chunks received when generating $y_i$. Thus, StreamST can be formulated as:

$$p(\mathbf{y} \mid \mathbf{s}, \mathbf{g}) = \prod_{i=1}^{I} p(y_i \mid \mathbf{s}_{\leq g_i}, \mathbf{y}_{<i}). \tag{1}$$

However, when the incoming speech stream becomes excessively long, StreamST models need to truncate historical speech and translations in real-time, thereby focusing on recent inputs while avoiding excessive inference latency (Iranzo-Sánchez et al., 2024). Consequently, truncation policy is employed to determine truncation timing. Let the truncation policy for the overall speech input and target translation be $\mathbf{a} = (a_1, ..., a_M)$ and $\mathbf{b} = (b_1, ..., b_M)$ respectively, where $M$ denotes the desired number of truncated segments, and $a_m$ and $b_m$ represent the ending positions of the $m$-th segment within the complete input stream and translation. Under the guidance of the segmentation policy, StreamST is reformulated as:

$$p(\mathbf{y} \mid \mathbf{s}, \mathbf{g}, \mathbf{a}, \mathbf{b}) = \prod_{i=1}^{b_1} p(y_i \mid \mathbf{s}_{1:g_i}, \mathbf{y}_{1:i-1}) \times \prod_{m=2}^{M} \prod_{i=b_{m-1}+1}^{b_m} p(y_i \mid \mathbf{s}_{a_{m-1}+1:g_i}, \mathbf{y}_{b_{m-1}+1:i-1}),$$

$$\tag{2}$$

where the streaming translation generation will be based solely on the input speech segment and output translation segment that remain after truncation. Therefore, StreamST requires determining both truncation and generation policies to guide the model in accomplishing translation generation.

**Chain-of-Thought Instruction** Chain-of-Thought (CoT) is originally developed for text-based tasks and has been proven to enhance performance on complex tasks by prompting large language models (LLMs) to think step by step before providing final results (Wei et al., 2022; DeepSeek-AI et al., 2025a). For speech inputs, CoT techniques have been widely adopted in speech-to-text

cross-modal tasks, where LSLMs first generate transcription and subsequently produce the final outputs (Zhang et al., 2023; Huang et al., 2023). In the context of speech translation, the model first generates transcription $\mathbf{x} = (x_1, ..., x_J)$, followed by the translation:

$$p(\mathbf{y} \mid \mathbf{s}) = p(\mathbf{y} \mid \mathbf{x}, \mathbf{s})p(\mathbf{x} \mid \mathbf{s}). \tag{3}$$

## 3 METHOD

In this section, we propose StreamUni, a framework that leverages speech CoT to consolidate all subtasks in StreamST. We begin by introducing the architecture of StreamUni and detailing its operational process for achieving StreamST. Subsequently, we present the truncation and generation policies within the StreamUni framework, which governs the management of historical speech and translation and the real-time translation generation. To further enhance the generation capabilities of LSLMs across multiple CoT stages under low-latency conditions, we propose a novel streaming CoT training scheme. The following subsections detail our methodology.

### 3.1 MODEL FRAMEWORK

The model framework of our approach is illustrated in Figure 1. StreamUni first transcribes the incoming speech input and compares the real-time transcription with the historical transcriptions to determine the truncation policy. If the truncation policy is triggered, StreamUni directly generates the translation bypassing the generation policy; otherwise, the number of words to be generated is determined by the generation policy. A more formalized operational process is presented as follows.

Given that the previous truncation timing of the input speech stream is $a_m$, and the current timing is $n$ $(n > a_m)$, the currently received speech segment fed into the model can be represented as $\mathbf{s}_{a_m+1:n}$. For segment $\mathbf{s}_{a_m+1:n}$, the LSLM first utilizes an audio encoder to encode it into speech embeddings. Following the speech CoT instruction, LSLM subsequently generates real-time transcription $\mathbf{x}^{(n)}$ of $\mathbf{s}_{a_m+1:n}$. StreamUni then determines the truncation policy by comparing $\mathbf{x}^{(n)}$ with maintained historical transcription queue, specifically deciding whether the current timing $n$ should trigger truncation. If it is determined that the current timing $n$ should trigger truncation, StreamUni disregards the generation policy, and continues generating and outputting all subsequent translation based on the input segment $\mathbf{s}_{a_m+1:n}$ and real-time transcription $\mathbf{x}^{(n)}$, building upon the already output translation segment $\mathbf{y}_{b_m+1:i-1}$, where $b_m$ is the translation truncation index corresponding to $a_m$. Otherwise, StreamUni determines the generation policy based on the real-time transcript $\mathbf{x}^{(n)}$ and uses it to determine the number of output words at current timing. We then elaborate on the truncation policy and generation policy in detail.

**Truncation Policy** StreamST employs a truncation policy to remove historical speech and translation segments no longer required for subsequent generation. To ensure truncation integrity, each truncated speech segment must maintain semantic alignment with its corresponding translation segment (Iranzo-Sánchez et al., 2024). The above truncation constraints serve dual purposes: (1) preventing the eliminated speech segment containing untranslated content, which would compromise generation quality, and (2) avoiding removal of already-translated content of remaining speech segment, which will result in repetitive translation of remaining speech segment. According to these, we propose the following truncation policy.

For speech stream $\mathbf{s} = (s_1, ..., s_N)$, StreamUni obtains real-time transcription after receiving each chunk and maintains a historical transcription queue $\mathbf{q}$. Assuming the end position of the previous truncated input segment is $a_m$ and the chunk to be processed is $n$ $(n > a_m)$, $\mathbf{q}$ can be represented as $= [\mathbf{x}^{(a_m+1)}, ..., \mathbf{x}^{(n-1)}]$. StreamUni first obtains the transcription $\mathbf{x}^{(n)}$ based on $\mathbf{s}_{a_m+1:n}$:

$$\mathbf{x}^{(n)} = \arg\max_{\mathbf{x}} p(\mathbf{x} \mid \mathbf{s}_{a_m+1:n}). \tag{4}$$

Subsequently, we compare $\mathbf{x}^{(n)}$ with items in $\mathbf{q}$ to determine the truncation policy. Speech segment truncation occurs if either condition is satisfied:

- If $\mathbf{x}^{(n)}$ remains identical to real-time transcriptions from the previous two chunks ($\mathbf{x}^{(n-1)}$ and $\mathbf{x}^{(n-2)}$), then $a_{m+1} = n$ becomes the speech truncation timing and $\mathbf{s}_{a_m+1:a_{m+1}}$ is discarded. The historical transcription queue is cleared ($\mathbf{q} = [\ ]$).

- If $\mathbf{x}^{(l)}(l = n-1, n-2)$ forms a complete sentence terminated by punctuation (.?!;), and $\mathbf{x}^{(n)}$ begins a new sentence following the complete sentence, then truncation timing is $a_{m+1} = l$ and $\mathbf{s}_{a_m+1:a_{m+1}}$ is discarded. The historical transcription queue is cleared, and the newly generated $\mathbf{x}^{(l)}(l = a_{m+1}+1, .., n)$ are sequentially added.

After determining the truncation timing, StreamUni generates and outputs the complete translation corresponding to $\mathbf{s}_{a_m+1,a_{m+1}}$ based on previously output translation:

$$\mathbf{y}_{i:b_{m+1}} = \arg\max_{\mathbf{y}} p(\mathbf{y} \mid \mathbf{s}_{a_m+1:a_{m+1}}, \mathbf{x}^{(a_{m+1})}, y_{b_m+1:i-1}), \tag{5}$$

where $b_{m+1}$ is the index of the last word in the output translation. The translation segment $\mathbf{y}_{b_m+1:b_{m+1}}$ is discarded.

In conclusion, we select truncation timing when users maintain prolonged silence or finish a full sentence, as content prior to this timing is relatively complete and subsequent translations are unlikely to reference earlier inputs. After determining input truncation timing, target truncation timing is decided by outputting complete translation for the truncated input segment, thereby maintaining semantic integrity of the truncation. More explanation is in Appendix A.

**Generation Policy** After establishing the truncation policy, we then determine the generation policy, which controls model output at all timing except truncation moments. The generation policy follows two key principles. First, the model should continue generating translation upon detecting the text within input speech; otherwise, no generation is required (Dong et al., 2022). Second, translation generation should lag behind the input source text to provide sufficient context for translation (Liu et al., 2021). Leveraging speech CoT, we implement the generation policy in Figure 1.

Assume the previous truncated segment is the $m$-th segment, and the speech chunk to be processed is $s_n$. We can obtain the transcription $\mathbf{x}^{(n)}$ using Eq.(4). Let $C$ denote the number of words in $\mathbf{x}^{(n)}$ and $i-1$ represent the position of the last word in the already output translation. The number of translation words allowed to be output is:

$$O = C - k - (i - 1 - b_m), \tag{6}$$

where the second term $k$ is the delay hyperparameter, and the third term represents the number of retained output translation words. This setting ensures that translation generation consistently lags behind the input text by $k$ words, providing sufficient context for generation. The current translation generation can be represented as:

$$\mathbf{y}_{i:i-1+O} = \arg\max_{\mathbf{y}} p(\mathbf{y} \mid \mathbf{s}_{a_m+1:n}, \mathbf{x}^{(n)}, y_{b_m+1:i-1}). \tag{7}$$

Then the generated translation $\mathbf{y}_{i:i-1+O}$ will be output.

## 3.2 Streaming CoT Training

After introducing the overall model framework, StreamUni can now perform StreamST using existing LSLMs (Microsoft et al., 2025; Xu et al., 2025). However, existing LSLMs are trained on multi-task datasets containing complete speech inputs paired with corresponding responses. In streaming scenarios with continuously growing speech stream, LSLMs must handle speech inputs of different lengths, which we refer to as streaming generation capability. Furthermore, our approach unifies policy decisions and streaming translation generation through speech CoT, which requires enhanced streaming generation capability across multiple stages of speech CoT. Therefore, we propose the Streaming CoT training scheme, which improves the capabilities of policy decision and streaming translation generation by augmenting streaming speech CoT data.

Our method constructs streaming CoT data using existing non-streaming CoT triplets of speech, transcription, and translation. Given the input speech stream $\mathbf{s} = (s_1, ..., s_N)$, our approach randomly truncates the stream through uniform sampling to obtain $\mathbf{s}_{\leq i}$. We then employ timestamp alignment tools to extract the corresponding transcription $\mathbf{x}^{(i)}$ for $\mathbf{s}_{\leq i}$ from the complete transcription $\mathbf{x}$. Our Streaming CoT training encourages the LSLM to predict full translation based on partial speech and transcription:

$$\mathcal{L} = - \sum_{\mathbf{s}_{\leq i} \sim \mathcal{U}(\mathbf{S})} \log p(\mathbf{y} \mid \mathbf{x}^{(i)}, \mathbf{s}_{\leq i}) \, p(\mathbf{x}^{(i)} \mid \mathbf{s}_{\leq i}), \tag{8}$$

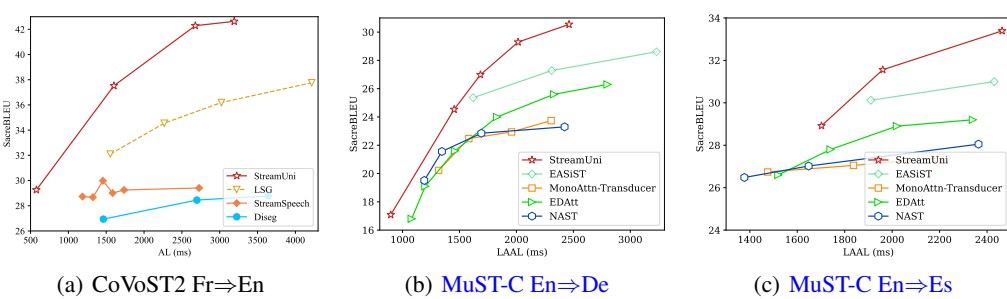

(a) CoVoST2 Fr⇒En       (b) MuST-C En⇒De       (c) MuST-C En⇒Es

Figure 2: Performance of different methods on SimulST task.

where $\mathbf{S}$ is $\{\mathbf{s}_{\leq 1}, ..., \mathbf{s}_{\leq N}\}$ and $\mathbf{s}_{\leq i} \sim \mathcal{U}(\mathbf{S})$ represents uniform sampling from set $\mathbf{S}$. This formulation trains accurate transcription prediction for policy decisions while requiring complete translation prediction to enhance generation capability and prevent premature termination. For efficiency, we employ sampling rather than training on all possible speech inputs for an instance. Through this training approach, our method efficiently enhance streaming CoT generation capability, thereby improving the capabilities of policy decision and streaming translation generation in low latency. In experiments, our training method requires integration with traditional non-streaming training approaches to achieve greater performance gains.

# 4 EXPERIMENTS

## 4.1 DATASETS

We mainly conduct experiments on streaming speech translation (StreamST) and simultaneous machine translation (SimulST) tasks.

**MuST-C English⇒German (En⇒De)** This dataset (Di Gangi et al., 2019) is collected from TED talks. The dataset contains both document-level and human-annotated sentence-level speech translation data, enabling evaluation of both SimulST and StreamST tasks.

**MuST-C English⇒Spanish (En⇒Es)** The dataset is constructed following the same approach as MuST-C En-De and serves as an evaluation benchmark for both StreamST and SimulST tasks.

**CoVoST2 English⇒Chinese (En⇒Zh)** This dataset only contains sentence-level speech translation data and is used to evaluate SimulST tasks (Wang et al., 2020).

**CoVoST2 French⇒English (Fr⇒En)** This dataset is also used to evaluate SimulST tasks.

## 4.2 SYSTEM SETTINGS

In this subsection, we delineate the settings of our StreamUni method and then present the comparative methods for each task separately.

For our approach, we adopt Phi-4-Multimodal (Microsoft et al., 2025) as the primary backbone LSLM and fine-tune it using the speech CoT data across four language directions. Specifically, the En⇒Zh direction contains 50 hours of streaming CoT data and 50 hours of non-streaming CoT data, while the other three directions each comprise 100 hours of non-streaming CoT data. The CoT instruction used for LSLM inference is: 'Transcribe the audio to text, and then translate the audio to **{target_lang}**. Use <sep> as a separator between the original transcript and the translation''. During inference, the chunk size is set to 320ms for the En-Zh direction and 640ms for the other directions. To control inference latency, we configure $k$ as $\{1, 3, 5, 7, 9\}$. When applied to the SimulST task, StreamUni executes only the generation policy. **Additional training hyperparameters are provided in the Appendix B**. Beyond Phi-4-Multimodal, we also experiment with Qwen2.5-Omni (Xu et al., 2025) as the base LSLM to validate the generalizability of our method, leveraging its thinker for policy-decision and translation generation.

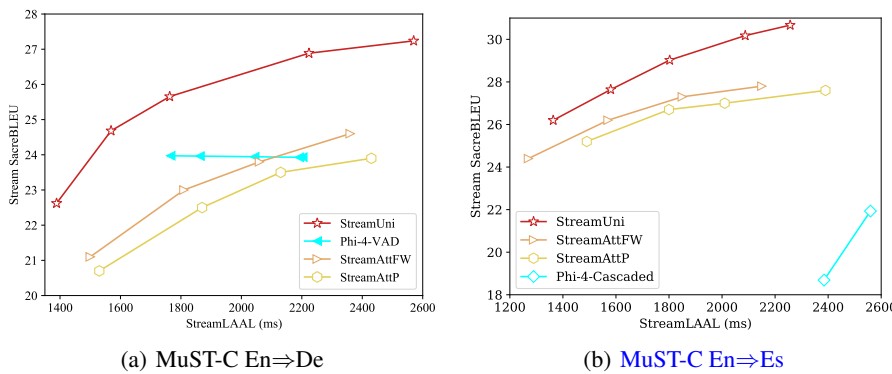

Figure 3: StreamST performance of different methods.

For SimulST task, we compare our method with encoder-decoder **DiSeg** (Zhang & Feng, 2023), **NAST** (Ma et al., 2023), **EDAtt** (Papi et al., 2023a), **StreamSpeech** (Zhang et al., 2024b), **LSG** (Guo et al., 2025), **MonoAttn-Transducer** (Ma et al., 2025) and **EASiST** (Fu et al., 2025). We also design a baseline called **Phi4-Wait-$k$**, which also uses fine-tuned Phi-4-Multimodal as our StreamUni but employs a generation policy that waits for $k-1$ chunks and then outputs one word for each subsequently received chunk.

For the StreamST task, we compare our method with **StreamAttFW** and **StreamAttP** (Papi et al., 2024). Furthermore, we implement a baseline called **Phi-4-VAD**, which replaces our truncation policy with VAD (Team, 2024) while keeping all other components consistent with our approach. In addition, we propose an additional cascaded method named **Phi-4-Cascaded**: we adopt Whisper-Large-V3 (Radford et al., 2022) as the ASR model and feed its outputs into Phi-4-Mini-Instruct for translation. The prompt used for Phi-4-Mini-Instruct is: "Translate the English text to German based on the given German translation. English text: {cot_asr}. German translation: {cot_st}".

### 4.3 EVALUATION

In evaluating streaming generation systems, we need to assess two critical aspects: latency and generation quality. To quantify latency, we utilize the Average Lagging (AL) (Ma et al., 2019) and Length-Adaptive Average Lagging (LAAL) metrics (Papi et al., 2022b), which measures the delay between input reception and output generation. For translation quality, we use the SacreBLEU (Post, 2018) and COMET (Rei et al., 2022) metrics. For the SimulST task, we employ the SimulEval tool (Ma et al., 2020a) to evaluate our StreamUni. In the StreamST task, we follow the setup of Papi et al. (2024). We first use mWERSegmenter (Matusov et al., 2005) for aligning document-level translation with references and then convert these alignments into consistent metrics used in the SimulST task. In this task, The latency metric is termed StreamLAAL, and translation quality is evaluated using Stream SacreBLEU by comparing the segmented document-level translations with the reference translations.

### 4.4 MAIN RESULTS

We evaluate our methods on SimulST and StreamST tasks.

As illustrated in Figure 2, our method achieves optimal SimulST performance across all datasets. Compared to traditional SimulST approaches employing Encoder-Decoder architectures (e.g., NAST and EDAtt), our method harnesses the comprehension and reasoning capabilities of LSLMs (Microsoft et al., 2025), yielding substantial performance improvements across all latency settings. Although methods like LSG also leverage LSLMs and demonstrate promising results, their policy decisions rely on heuristic rules (Guo et al., 2025), resulting in suboptimal performance. Furthermore, compared to EASiST built on larger-scale foundation models as backbones and trained with customized policy data (Fu et al., 2025), our model achieves equally better performance. This is mainly attributed to our method's utilization of intermediate outputs from speech CoT, which enables real-time detection of valid user inputs and allows generation decisions to be made at optimal

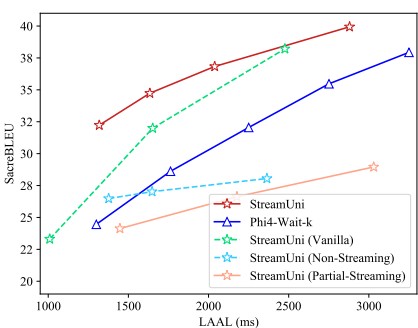 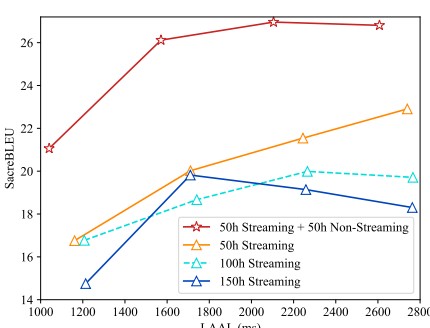

Figure 4: The SimulST performance on En⇒Zh with different policies and training methods: Vanilla (no extra training), Non-Streaming (trained solely on non-streaming data), and Partial-Streaming ( encourages the model to predict the corresponding translation based on the partial speech input, rather than translation corresponding to the unsegmented speech input.

Figure 5: The SimulST performance of different training data recipes on MuST-C En⇒De dataset. Our method utilizes 50 hours each of proposed streaming and non-streaming data, whereas other methods only employ the proposed streaming data with varying total durations.

timings. Through our superior policy and streaming CoT training scheme, we achieve further performance gains.

Our method also demonstrates superior performance on the StreamST task, as shown in Figure 3. Traditional StreamST approaches, including StreamFW and StreamAttP (Papi et al., 2024), rely on attention interpretability to determine generation and truncation policies for streaming translation. In contrast, our approach utilizes speech CoT for real-time detection of valid speech inputs to inform generation policies, while implementing truncation policies through alignments between speech input and translations. This design enables more effective policy decisions and enhanced performance. Compared to Phi-4-VAD, which employs VAD for truncation policy, our method achieves truncation policy through the semantic alignments between speech inputs and translation, resulting in more appropriate timing and enhanced performance. Relative to Phi-4-Cascaded, our method delivers a more substantial performance boost—particularly in terms of latency—highlighting the significant advantages of end-to-end models over cascaded counterparts. Notably, we anticipate that Phi-4-Mini-Instruct will achieve further improvements with dedicated training.

In addition to the above results, we also considered the usability of our method. To this end, we additionally incorporated the actual machine inference time into the latency metric when calculating latency, which is denoted as Computation-Aware LAAL (Xu et al., 2024). It essentially measures the average delay from the user's speech input to the machine's output of the corresponding translation. For details, refer to Appendix E.

## 5 ANALYSIS

To provide deeper understandings into our approach, we conduct comprehensive analyses, with each experiment detailed below.

### 5.1 ABLATION STUDY

We first conduct ablation studies to investigate the impact of different configurations.

Figure 4 presents a performance comparison of our method under various training methods and generation policies. Unlike Phi4-Wait-$k$, which employs heuristic rules for generation decisions without considering speech content (Ma et al., 2019), our method determines generation timing by detecting valid speech inputs, thereby achieving superior performance through more informed generation policies. Beyond generation policy, our proposed streaming CoT training scheme enhances

performance across all latency settings, particularly under low-latency. However, the streaming CoT training data must be combined with non-streaming data to achieve maximum performance gains.

To validate this hypothesis, we conduct experiments using training data from a single language direction. As illustrated in Figure 5, simply increasing data volume when using only streaming CoT data fails to yield performance improvements. Superior performance across different latency settings is achieved exclusively when both streaming and non-streaming CoT data are employed simultaneously. We hypothesize that this mixed-data approach effectively stimulates the streaming generation capabilities of model while enabling it to perceive complete speech input boundaries, thereby preventing over-translation and achieving enhanced overall performance.

Furthermore, we investigate the effectiveness of our proposed truncation policy. Rather than comparing the accuracy of model-determined truncation timing against official truncation points, we focus on the final generation quality, which represents our ultimate objective. Given document-level speech inputs with an average duration exceeding 10 minutes (Di Gangi et al., 2019), we evaluate the generation quality of our fine-tuned model under different truncation policies. Table 1 illustrates the results, where we report document-level metrics rather than sentence-level metrics after alignment. Notably, while our proposed truncation strategy performs slightly below the human-annotated policy on SacreBLEU, it surpasses official annotation on COMET score. This demonstrates the effectiveness of our approach and provides valuable insights for future research utilizing semantic alignment models to implement truncation policies. We also provide an analysis in Appendix F of why our method, despite a slightly lower SacreBLEU, outperforms the officially provided truncation policy in terms of COMET.

Table 1: Performance on MuST-C with document-level speech inputs when the generation policy is disabled and only truncation is employed. 'Human' uses annotated truncation timing, while 'Model' lets the system decide automatically.

| Direction | Truncation | COMET | SacreBLEU |
|-----------|------------|-------|-----------|
| **En⇒De** | official | 82.45 | **32.51** |
|           | Model | **83.42** | 31.59 |
| **En⇒Es** | official | 80.83 | **35.84** |
|           | Model | **82.86** | 34.97 |

Furthermore, we also demonstrate the impact of adopting Speech CoT on translation performance. Herein, we use the RealSI dataset (Cheng et al., 2024) to explore full-sentence speech translation performance with and without Speech CoT. Please refer to Appendix D for more details. We find that Speech CoT significantly enhances speech translation performance in real-world scenarios.

## 5.2 SPEECH CoT ARGUMENTATION

StreamUni unifies streaming translation generation and policy decisions through speech CoT. The accuracy of outputs at each CoT stage significantly impacts overall StreamST performance, particularly under low-latency settings. To investigate this, we construct a low-latency speech evaluation dataset based on CoVoST2 En⇒Zh to assess generation capabilities across different CoT stages.

For dataset construction, we randomly truncate speech clip and obtain transcriptions using WhisperX (Bain et al., 2023), then generate reference translations using the DeepSeek-V3-0324 model (DeepSeek-AI et al., 2025b). We evaluate models trained with different schemes through speech CoT inference. More details are in Appendix C. As shown in Table 4, our approach achieves superior performance across all CoT stages, delivering excellent capabilities of policy decision and streaming translation generation.

## 5.3 EXTENDING TO OTHER LSLMs

Beyond the analytical experiments of our method, we further extend our evaluation to Qwen2.5-Omni-7B (Xu et al., 2025) to validate the generalizability of our approach across different LSLMs. The experimental results are presented in Table 2. Phi-4-Multimodal consistently outperforms Qwen-Omni on both ST and SimulST tasks, demonstrating that LSLMs with stronger speech translation capabilities achieve superior SimulST performance. This finding further validates that our StreamUni method can effectively leverage and scale with the enhanced capabilities of LSLMs, thereby demonstrating the generalizability of our approach.

Table 2: Performance of various vanilla LSLMs on ST and SimulST tasks. 'ST' denotes speech translation that utilizes complete speech inputs for translation, while 'SimulST' represents the simultaneous speech translation task that incorporates our proposed generation policy. The evaluation dataset is the MuST-C En⇒De sentence-level dataset.

| Task | Base Model | LAAL(↓) | SacreBLEU(↑) |
|---|---|---|---|
| **ST** | Phi-4-Multimodal | N/A | 28.55 |
| | Qwen2.5-Omni | N/A | 24.21 |
| **SimulST** | Phi-4-Multimodal | 1112.48 | 22.51 |
| | | 1448.43 | 24.27 |
| | Qwen2.5-Omni | 949.36 | 20.64 |
| | | 1449.83 | 21.80 |

## 6 RELATED WORK

Streaming speech translation (StreamST) aims to generate real-time translations for continuously arriving speech stream, requiring the simultaneous completion of generation policy, segmentation policy, and streaming translation generation. Early research focused on sentence-level speech segments and is called simultaneous speech translation (SimulST), predominantly employing encoder-decoder architectures (Vaswani et al., 2017). Initial SimulST methods (Ma et al., 2020c) determine generation policy based on the number of input chunks. Subsequently, researchers explore content-adaptive generation policy by leveraging auxiliary ASR tasks (Zeng et al., 2021; Chen et al., 2021; Zhang et al., 2024b), integrate-and-fire (Dong et al., 2022), monotonic attention (Communication et al., 2023), transducer (Liu et al., 2021; Tang et al., 2023), and CTC (Graves et al., 2006; Ma et al., 2023) to make decisions based on speech content. At the same time, some methods (Weller et al., 2021; Papi et al., 2023b; Omachi et al., 2022) attempt to accomplish translation by continuously refreshing the output translations.

With the advancement of Large Speech-Language Models (LSLMs), researchers have begun exploring their application to SimulST tasks (Agostinelli et al., 2024; Guo et al., 2025; Fu et al., 2025). Hibiki (Labiausse et al., 2025) even achieves simultaneous speech-to-speech translation in an end-to-end manner. However, relying solely on LSLMs for SimulST still requires coordination with multiple auxiliary models to achieve complete StreamST, introducing cascaded errors and hindering end-to-end optimization (Li et al., 2021). Consequently, researchers have attempted to develop unified methods capable of handling all StreamST tasks within a single model framework. Early attempts utilize attention mechanisms for generation and segmentation decisions (Papi et al., 2024), while subsequent work constructs dedicated policy-specific datasets to enable autoregressive prediction for policy decisions (Cheng et al., 2024; Ouyang et al., 2025). Nevertheless, these approaches suffer from significant challenges in large-scale data construction and advanced model transferability, while facing difficulties in fully leveraging the pre-training capabilities of foundation models.

## 7 CONCLUSION

In this paper, we propose StreamUni, a framework that efficiently enables unified LSLM to accomplish all subtasks of StreamST in a cohesive manner. By unifying different subtasks formats into autoregressive generation, StreamUni can achieve streaming translation with only a small amount of streaming CoT training data. Experiments show that our method efficiently attains state-of-the-art performance on StreamST tasks across multiple language directions with the same volume of training data. Furthermore, analytical experiments verify the effectiveness of each module in StreamUni as well as its practical usability.

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

Table 3: Settings of StreamUni.

| Hyperparameters | | | Settings |
|---|---|---|---|
| | Base_model | Base_model | Phi-4-Multimodal |
| LSLM | Training Details | batch_size | 32 |
| | | learning_rate | 4e-5 |
| | | weight_decay | 0.01 |
| | | lr_scheduler | WarmupLR |
| | | betas | (0.9, 0.95) |
| | | optimizer | AdamW |
| | | zero_optimization | stage_2 |

Shaolei Zhang, Qingkai Fang, Shoutao Guo, Zhengrui Ma, Min Zhang, and Yang Feng. Stream-Speech: Simultaneous speech-to-speech translation with multi-task learning. In Lun-Wei Ku, Andre Martins, and Vivek Srikumar (eds.), *Proceedings of the 62nd Annual Meeting of the Association for Computational Linguistics (Volume 1: Long Papers)*, pp. 8964–8986, Bangkok, Thailand, August 2024a. Association for Computational Linguistics. doi: 10.18653/v1/2024.acl-long.485. URL https://aclanthology.org/2024.acl-long.485/.

Shaolei Zhang, Qingkai Fang, Shoutao Guo, Zhengrui Ma, Min Zhang, and Yang Feng. Stream-Speech: Simultaneous speech-to-speech translation with multi-task learning. In *Proceedings of the 62nd Annual Meeting of the Association for Computational Linguistics*, August 2024b.

## A  TRUNCATION POLICY IN STREAMUNI

Our truncation policy is designed to truncate historical speech inputs and translations in real-time, enabling the model to focus on recent speech inputs while avoiding additional inference costs. To this end, we establish two key principles: (1) preventing elimination of speech segments containing untranslated content, which would compromise generation quality, and (2) avoiding removal of already-translated content from remaining speech segments, which would result in repetitive translations. Our approach first identifies the truncation timing for source speech inputs, then uses this as an anchor to determine the corresponding truncation point for output translations.

For speech inputs, we consider appropriate truncation timing to be when users pause speaking or complete a sentence. Therefore, we design two triggering rules for speech truncation. The first rule targets prolonged user silence, while the second targets moments when users finish speaking a complete sentence. When neither condition is met for an extended period, causing the processed speech duration to exceed a predefined threshold (30 seconds), our method designates this moment as the truncation point.

After determining the speech truncation timing, we identify the corresponding truncation point for target translations to ensure semantic consistency between discarded content on both source and target sides. To achieve this alignment, we instruct the model to output all translations preceding the source truncation point and subsequently discard them.

This approach implements an effective truncation policy that maintains translation quality while ensuring computational efficiency.

## B  TRAINING AND EVALUATION DETAILS

We provide comprehensive details of our training methodology. For training data construction, we focus on building streaming CoT data for the En⇒Zh direction and incorporate an equal duration of non-streaming CoT data. For other language pairs, we directly utilize non-streaming data. The dataset released in this work is intended for academic research purposes only. Any commercial use is strictly prohibited. Our training details are detailed in Table 3.

Table 4: Performance of multiple stages of speech CoT under different training configurations. 'Streaming CoT + Non-Streaming CoT' denotes our employed training recipe. 'Non-Streaming CoT' only utilizes Non-Streaming CoT training data. 'Vanilla' represents the baseline without any further training.

| Training Settings | WER ($\downarrow$) | SacreBLEU ($\uparrow$) |
|---|---|---|
| Streaming CoT + Non-Streaming CoT | **20.74** | **35.22** |
| Non-Streaming CoT | 27.83 | 33.60 |
| Vanilla | 31.62 | 33.34 |

Table 5: Full-Sentence Translation Performance: With vs. Without Speech CoT Argumentation.

| Settings | SacreBLEU |
|---|---|
| Direct Trans | 22.08 |
| CoT Trans | 25.23 |

For SimulST evaluation, we employ SimulEval (Ma et al., 2020a) as the standard assessment framework. For StreamST evaluation, we first utilize mWERSegmenter (Matusov et al., 2005) alignment tools to map the generated document-level translations to sentence-level references. Subsequently, we compute latency metrics and translation quality on the aligned sentences. We refer to these metrics as Stream LAAL (Papi et al., 2024) and Stream SacreBLEU, respectively.

## C  SPEECH COT ARGUMENTATION

For streaming speech translation, the key challenge lies in real-time performance. This challenge is amplified under extremely low latency, where very short input segments make accurate translation and policy decision-making especially important. To evaluate these aspects, we construct a dedicated Low-Latency Speech Evaluation dataset. This evaluation set was derived from conventional speech translation corpora (consisting of speech segments, transcripts, and translations) through the following modifications:

- For each complete speech segment, we randomly sample speech prefixes with shorter duration based on the given complete speech segment.

- Using WhisperX (Bain et al., 2023), we obtain word-level timestamps for the complete speech segment, which allows us to extract the corresponding ground-truth transcript prefixes for the sampled speech prefixes.

- With DeepSeek-V3-0324 (DeepSeek-AI et al., 2025b), we generate ground-truth translations of the transcript prefixes, yielding the low-latency speech evaluation dataset.

During evaluation, the model is encouraged to generate intermediate results of speech CoT based on speech prefixes at different stages. For the ASR transcription outputs of speech CoT, we compute the WER against the ground-truth transcription prefixes to assess its low-latency transcription capability, which further reflects its policy-decision ability under low latency for our StreamUni. For the translation results of speech CoT, we calculate SacreBLEU against the ground-truth references to measure its low-latency translation capability. Based on the constructed evaluation dataset, We evaluate models trained with different schemes through speech CoT inference.

The detailed experimental results are shown in Table 4. Our employed 'Streaming CoT + Non-Streaming CoT' training scheme achieves lower WER and higher SacreBLEU scores, our approach achieves superior performance across all CoT stages, delivering excellent capabilities of policy decision and streaming translation generation.

Table 6: Computation-Aware latency and translation results on MuST-C En⇒De task.

| Computation-Aware Stream LAAL (ms) | Stream LAAL (ms) | Stream SacreBLEU |
|:---:|:---:|:---:|
| 3193.94 | 1762.85 | 25.65 |
| 3690.02 | 2223.69 | 26.89 |

Table 7: Computation-Aware latency and translation results on MuST-C En⇒Es task.

| Computation-Aware Stream LAAL (ms) | Stream LAAL (ms) | Stream SacreBLEU |
|:---:|:---:|:---:|
| 3077.07 | 1802.59 | 29.02 |
| 3405.03 | 2087.39 | 30.18 |

## D  FULL-SENTENCE TRANSLATION PERFORMANCE WITH SPEECH COT ARGUMENTATION

In this section, we compare the performance of direct translation and CoT translation with on the full-sentence speech translation task. Here, we use RealSI (Cheng et al., 2024), a speech test set from real-world scenarios containing English⇒Chinese directions. It can be observed in Table 5 that CoT also brings significant improvements in translation performance.

## E  COMPUTATION-AWARE LATENCY

To validate the feasibility of our method for real-world deployment, we explicitly incorporate computational latency into our evaluation framework when conducting experiments on the NVIDIA GeForce RTX 3090 (a consumer-grade GPU). Specifically, we adopt the computation-aware LAAL metric (Papi et al., 2022a; Xu et al., 2024), which quantifies the end-to-end latency from the user's speech input to the model's translation output. The table below reports our method's StreamST performance on the MuST-C En⇒De and En⇒Es datasets, with both computation-aware latency and non-computation-sensitive metrics included for comprehensive comparison.

As can be seen from the Table 6 and Table 7, our method achieves promising performance with a latency of approximately 3 seconds when computational costs are taken into account, and around 2 seconds when computational delays are not considered. Notably, these results are obtained without leveraging any inference optimization frameworks or advanced GPUs. It is anticipated that the adoption of the aforementioned optimization techniques will enable us to achieve even much lower latency.

## F  ANALYSIS ON THE PATTERN OF DIFFERENT TRUNCATION POLICIES

To explore Why the ours truncation strategy outperforms MuST-C's official truncation policy in terms of COMET scores, we conduct a detailed analysis of the average duration of processed speech inputs and average translation length obtained from two truncation policies. The results are shown in the Table 8.

We find that the truncation policy of our method enables the model to process longer speech segments and reference lengthier historically generated translations. This provides sufficient context

Table 8: Analysis of input and output patterns of different truncation policies on MuST-C En⇒De task.

| Settings | Avg Duration | Avg Translation Length |
|:---:|:---:|:---:|
| StreamUni | 9.34 | 27.91 |
| MuST-C | 5.77 | 15.6 |

for the model during translation, thus ensuring that our truncation policy outperforms the truncation approach of MuST-C.

