# OpenReview forum: "StreamUni: Achieving Streaming Speech Translation with a Unified Large Speech-Language Model"
_ICLR.cc/2026/Conference — ICLR 2026 Conference Withdrawn Submission_

### Official Review · Reviewer_hNPi · 2025-10-19

**Soundness:** 2
**Presentation:** 2
**Contribution:** 2
**Rating:** 2
**Confidence:** 5

**Summary:**

In this work, the authors propose a streaming speech translation system based on large speech language model. The system includes a truncation policy to keep the input audio length in a reasonable number and suitable for streaming translation task last for hours. The ASR transcription
is also included to the language model output, i.e. COT, to boost the performance. The final results are very competitive. However, my main concern is the novelty of this work. Combining both speech recognition and translation results for streaming application has been proposed before [1]; The generation policy is based on wait-k, which has already shown been suboptimal in many streaming translation papers, such as [2]; Also, the comparison in the experiment is unfair and it is hard to figure out the main factor contributing to the good performance.

[1] Weller, Orion et al. “Streaming Models for Joint Speech Recognition and Translation.” EACL (2021).
[2] Papi, Sara et al. “Attention as a Guide for Simultaneous Speech Translation.” ACL (2023).

**Strengths:**

- A good streaming system to leverage SOTA LLM for streaming speech translation.
- A truncation policy to limit the input audio length.

**Weaknesses:**

- COT proposed has been studied before as discussed in the summary section
- wait-k based generation policy is suboptimal.
- the comparison in the experiment is unfair and it is hard to figure out the main factor contributing to the good performance.  See Question section for more details.
- Forcealignment is required to train the model
- Truncated speech input/text output, especially text output,  leads to context information loss. It actually degrade the documentation level translation to utterance level based translation.

**Questions:**

- Not sure the main difference between SimulST and StreamST. Methods like Transducer, CAAT and TAED can be used for streaming transcription or translation for audio last for hours too, though the results reported in the literature are utterance based.
- How to add incremental speech chunk inputs during inference? Do we re-evaluated everything, from truncation policy, ASR outputs, to translation results?
- How do you determine the end of transcription text token generation give $x_n$? Will the LLM generate end of sentence token for the partial sentence.
- The comparison in the experiment is unfair and we don't know which factor contributing most to the good performance. LLM, COT or something else?
- What's the offline translation results from Phi-4-Multimodal? Does it leverage ASR results?
- What's the WER for the transcription generated?
- Why Phi-4-Multimodal - VAD is much worse than the proposed system? The proposed system will detect utterance boundary (Truncation time) by comparing decoding results from 3 consecutive chunks. Assume each chunk is 320ms, the silence segment should be around 640ms to 960ms. It should not be hard to for the VAD system to detect that.
- Which dataset is used in Table 2?

---

> ### Author Response · Authors · 2025-11-23
> **Response (1/4)**
>
> **Thank you very much for your helpful suggestions.**
>
> >COT proposed has been studied before in [1]?
>
> We greatly appreciate you bringing this work to our attention and providing the reference paper, which has been incorporated into our manuscript as a citation. However, we have identify **significant distinctions between our work and the referenced study, primarily in the following aspects**:
>
> (1) **Motivation**: The work in [1] adopts an interleaved approach for the model to alternately generate transcriptions and translations. **Its core objective is to provide transcriptions as supplementary information to users, with no application in core policy-decision or translation quality improvement**. In contrast, our method leverages ASR outputs within CoT to implement truncation policies and simultaneous interpretation policies. **Our primary goal is to efficiently equip speech LLMs with long-duration real-time simultaneous translation capabilities**.
>
> (2) **Method**: The model in [1] outputs ASR text and translations in an interleaved manner without implementing specific policies. **Instead, it continuously refreshes historical outputs by updating translations, which may not explore addressing real-world real-time translation challenges**. Our approach unifies all tasks required for long-duration real-time simultaneous translation—including truncation policy, generation policy, and translation generation—through Speech CoT. **This design enables us to achieve excellent long-duration simultaneous translation performance with minimal training on streaming CoT data**.
>
> (3) **Results**: **The work in [1] fails to demonstrate the advantages of the interleaved generation scheme. Notably, Table 1 even presents a counterintuitive phenomenon: translation quality is higher when translations are generated prior to ASR text.** In contrast, **our method not only showcases the superiority of Speech CoT in generation policies and truncation policies but also highlights its positive contribution to the final translation quality**.
>
> [1] Weller et al. Streaming Models for Joint Speech Recognition and Translation. EACL 2021.
>
> >Wait-k based generation policy is suboptimal?
>
> Your insights on this issue are valuable. The core of StreamUni lies in the integration of truncation policies, generation policies, and real-time translation through speech CoT. This design enables strong long-duration simultaneous translation performance using only streaming CoT data.
>
> When determining the generation policy, we apply a wait-k policy between the recognized ASR text and the target translation. However, **the generation policy is dynamic when viewed from the perspective of input speech to generated translation**. Unlike approaches such as [1], our method evaluates whether to produce a translation only when valid speech input is received, thereby accounting for the actual content of the speech signal. As a result, a dynamic policy is established between the input speech and the target translation, with core ideas closely aligned with previous sentence-level dynamic generation policy [2].
>
> We are confident that as more open-source generation policy data becomes available, our generation policy can be further refined to achieve even stronger performance.
>
> [1] Ma et al. SimulMT to SimulST: Adapting Simultaneous Text Translation to  End-to-End Simultaneous Speech Translation. AACL 2020.\
> [2] Dong et al. Learning When to Translate for Streaming Speech. ACL 2022.
>
> >The main difference between SimulST and StreamST?
>
> **The key difference between SimulST and StreamST lies in their ability to handle speech duration**. **SimulST generally only supports sentence-level speech, typically within 20 seconds**, where the model only needs to determine the output timing of the target translation—i.e., the generation policy. **In contrast, StreamST aims to process document-level speech, with durations ranging from tens of minutes to several hours**. Due to the longer input sequences, StreamST needs to not only decide the generation policy but also implement a truncation policy. This real-time truncation of historical speech inputs and previously generated translations helps avoid compromising inference efficiency.
>
> Methods such as Transducer, CAAT, and TAED primarily focus on determining the generation policy. **While theoretically capable of handling unlimited-length speech, they face practical challenges**. On one hand, their high processing latency and computational costs hinder real-world implementation. On the other hand, there is a lack of large-scale document-level speech translation training data. More importantly, these methods are difficult to scale to speech LLMs, as their training complexity would increase by an order of magnitude compared to StreamUni.

---

> ### Author Response · Authors · 2025-11-23
> **Response (2/4)**
>
> >How to add incremental speech chunk inputs during inference? Do we re-evaluated everything, from truncation policy, ASR outputs, to translation results?
>
> The incremental speech chunk will be appended to the historical speech input, and then the speech LLMs will generate the ASR results for the current speech input based on this combined input speech. Subsequently, the current ASR result will be compared with the historical ASR results to determine the truncation policy. If truncation policy is decided, the model will continue generating the translation based on the retained historical output translation. Otherwise, the model will continue generating the translation based on the retained historical output translation in accordance with the generation policy.
>
> >How do you determine the end of transcription text token generation given $x_n$?
>
> We do not rely on special tokens to determine the end of the transcription text. This is because such an approach not only introduces judgment errors but also increases training complexity. Instead, we decide whether to truncate the ASR text by comparing the current ASR text with historical ASR texts, based on the following two conditions:
>
> (1) The ASR text remains consistent across multiple consecutive model steps, indicating the user has not spoken for an extended period and truncation is therefore appropriate.\
> (2) The model starts a new sentence at the current step, immediately following the completion of a sentence in the previous step—under this circumstance, the ASR text from the previous step should be truncated.
>
> The above process is described in detail in the truncation policy. We also conduct additional experiments to investigate the feasibility of the above conditions. **Our analytical experiments show that when the maximum truncation duration is constrained to 30 seconds, the model meets the above conditions in 89.11% of cases—rather than being forcibly truncated after reaching the maximum duration**. Thus, the ASR text is usually truncated before reaching 30 seconds, which validates the effectiveness of our method.

---

> ### Author Response · Authors · 2025-11-23
> **Response (3/4)**
>
> >The comparison in the experiment is unfair and we don't know which factor contributing most to the good performance. LLM, COT or something else?
>
> Due to the limitations of existing SimulST schemes and StreamST methods, achieving a fully fair comparison in this paper is challenging. However, **we provide the following explanations to validate the effectiveness of different modules in our approach**.
>
> (1) We first verify the effectiveness of our generation policy. For this, we compare our method with EASiST on the SimulST task, where EASiST adopts Llama3-8B as the base LLM. Experimental results demonstrate that our method achieves a performance improvement of over 1 BLEU on MuST-C En-De and MuST-C En-Es subtasks under the same latency. **This validates the effectiveness of the generation policy in our approach**.
>
> | MuST-C En-De | | | | |
> |:--:|:--:|:--:|:--:|:--:|
> | **StreamUni** | LAAL | 1683.09 | 1808.49 | 2350.03 |
> | | SacreBLEU | 26.98 | 28.00 | 30.55 |
> | **EASIST** | LAAL | 1620 | 2310 | 3230 |
> | | SacreBLEU | 25.36 | 27.28 | 28.61 |
>
> | MuST-C En-Es | | | | |
> |:--:|:--:|:--:|:--:|:--:|
> | **StreamUni** | LAAL | 1702.22 | 1960.39 | 2463.44 |
> | | SacreBLEU | 28.92 | 31.56 | 33.39 |
> | **EASIST** | LAAL | 1910 | 2430 | 3460 |
> | | SacreBLEU | 30.12 | 31.00 | 31.92 |
>
> (2) Subsequently, we validate the effectiveness of the truncation policy on the StreamST task. With all other settings kept consistent, we compare the truncation policy mechanism of StreamUni with the method using silero-vad, based on the same speech LLMs. The experimental results are shown in the following table. By incorporating semantic information, our method avoids the model simply truncating historical content based on whether a person is speaking. This not only prevents inappropriate decision-making timings but also yields better translation quality. **This validates the effectiveness of the truncation policy in our approach**.
>
> | MuST-C En-De | | | |
> |:--:|:--:|:--:|:--:|
> | **StreamUni** | Stream LAAL | 1762.85 | 2223.69 |
> | | Stream SacreBLEU | 25.65 | 26.87 |
> | **Phi-4-VAD** | Stream LAAL | 1774.12 | 2204.71 |
> | | Stream SacreBLEU | 24.44 | 24.37 |
>
>
> (3) Additionally, we compare the performance of direct translation and CoT translation with on the full-sentence speech translation task. Here, we use RealSI [2], a speech test set from real-world scenarios. It can be observed that CoT also brings significant improvements in translation performance. **This validates the effectiveness of the CoT translation in our approach**.
>
> | RealSI En-Zh | | | |
> |:--:|:--:|:--:|:--:|
> | **Settings** | **SacreBLEU** |
> | Direct Translation | 22.08 |
> | CoT Translation | 25.25 |
>
> **Overall, our method enhances the final translation performance through its generation policy, truncation policy, and CoT integration. However, due to the interdependencies between these modules, it is difficult to isolate the relative effectiveness of individual components. All the aforementioned results have been supplemented in the revised paper.**
>
> [1] Fu et al. Efficient and Adaptive Simultaneous Speech Translation with Fully Unidirectional Architecture.\
> [2] Cheng, et al. Towards Achieving Human Parity on End-to-end Simultaneous Speech Translation via LLM Agent.

---

> ### Author Response · Authors · 2025-11-23
> **Response (4/4)**
>
> >What's the offline translation results from Phi-4-Multimodal? Does it leverage ASR results?
>
> The performance of Phi-4-Multimodal after streaming CoT training on the full-sentence speech translation task is shown in the following table.
>
> | Offline Translation | | | |
> |:--:|:--:|:--:|:--:|
> | **Datasets** | **SacreBLEU** |
> | CoVoST2 Fr-En | 44.68 |
> | MuST-C En-De | 28.55 |
> | MuST-C En-Es | 33.43 |
>
> The above results leverage the intermediate ASR results.
>
> >What's the WER for the transcription generated?
>
> We test the ASR performance metric (WER) on the MuST-C En-Es task, achieving a value of 7.67. This is a relatively low error metric, which validates the effectiveness of our method.
>
> >Why Phi-4-Multimodal-VAD is much worse than StreamUni on the StreamST task?
>
> Your question is valuable. Through analysis, we find that although VAD often aligns with StreamUni in making truncation decisions, it fails to account for the speaker's semantics and triggers truncation excessively frequently. This reduces the contextual information available to the model during translation, ultimately leading to suboptimal translation performance.
>
> >Which dataset is used in Table 2?
>
> Table 2 utilizes the MuST-C En-De sentence-level dataset. We have supplemented this description in the title of Table 2.
>
> >Truncate speech input/text output, especially text output may degrade the documentation level translation?
>
> Your question is valuable. In this paper, we focus on enabling speehc LLMs to achieve long-duration simultaneous translation capabilities with low training costs. Expanding the models's context window to enhance performance in scenarios such as noun reference consistency is a reasonable direction, and we will explore this in our future work.
>
> **Regarding the above responses we provided to you, we have already added and revised the relevant content in the paper. If our responses meet your satisfaction, we would greatly appreciate it if you could consider raising the score.**

---

> ### Comment · Reviewer_hNPi · 2025-11-25
> **Response Comments**
>
> I appreciate authors detailed replies and adding more evidence to support this work.
>
> *Novelty*
>
> As mentioned in the review, interleaved decoding is widely used in many streaming speech translation studies besides Weller's work,  such as [1][2]. At that time, people might not call it as COT but the methods are the same proposed in this work.  Interleaved decoding is also widely used for e2e based speech dialogue systems such as [3][4]. More evidence is required to demonstrate the novelty in this work.
>
> *SimulST and StreamST*
>
> I am still not convinced that we need to introduce another term to streaming ST with hours long speech.
>
> *Methods such as Transducer, CAAT, and TAED primarily focus on determining the generation policy. While theoretically capable of handling unlimited-length speech, they face practical challenges. On one hand, their high processing latency and computational costs hinder real-world implementation.*
>
> I am not sure why models like Transducer, CAAT, and TAED are with high processing latency and computational cost. Are you referring to the training or inference? Also, those methods have already been explored with large scale dataset [5]
>
> [1] Papi, Sara et al. “Token-Level Serialized Output Training for Joint Streaming ASR and ST Leveraging Textual Alignments.” 2023 IEEE Automatic Speech Recognition and Understanding Workshop (ASRU) (2023)
>
> [2] Omachi, Motoi et al. “Align, Write, Re-Order: Explainable End-to-End Speech Translation via Operation Sequence Generation.” ICASSP(2022)
>
> [3] Nguyen, Tu et al. “SpiRit-LM: Interleaved Spoken and Written Language Model.” Transactions of the Association for Computational Linguistics 13 (2024)
>
> [4] Wu, Donghang et al. “Chronological Thinking in Full-Duplex Spoken Dialogue Language Models.” ArXiv abs/2510.05150 (2025)
>
> [5] Xue, Jian et al. “Large-Scale Streaming End-to-End Speech Translation with Neural Transducers.” Interspeech (2022).

---

> > ### Author Response · Authors · 2025-11-27
> > **Further Response to Reviewer hNPi (2/2)**
> >
> > Additionally, by unifying all tasks in simultaneous translation through CoT, **our proposed streaming CoT argumentation scheme can boost all capabilities required for simultaneous translation by strengthening the speech CoT ability using only sentence-level speech training data—demonstrating the effectiveness of our approach**.
> >
> > > Explanation of SimulST and StreamST
> >
> > In our method, we aim to extend the processing duration of speech simultaneous translation models from the traditional tens of seconds to tens of minutes or even hours. **To distinguish these two processing scenarios, we introduce "StreamST" to refer to long-duration streaming speech translation**.
> >
> > Methods such as Transducer, CAAT, and TAED all belong to the Transducer-based framework. Their training complexity involves a dynamic programming step with a complexity of O(U*T) [1], where U and T represent the lengths of the source and target sequences, respectively. **Thus, when dealing with speech inputs and outputs spanning hours, the excessive computational complexity and high memory requirements make model training infeasible**. **Although such methods have been validated on large-scale datasets [2], their effectiveness for long-duration speech inputs remains unproven**.
> >
> > **Our StreamUni model only requires training on sentence-level speech data to extend its simultaneous translation capabilities to long-duration scenarios—demonstrating the feasibility and effectiveness of our approach.**
> >
> > [1] Ma,et al. Overcoming Non-monotonicity in Transducer-based Streaming Generation. ICML 2025.\
> > [2] Xue, et al. Large-Scale Streaming End-to-End Speech Translation with Neural Transducers. InterSpeech 2022.

---

> ### Author Response · Authors · 2025-11-27
> **Further Response to Reviewer hNPi (1/2)**
>
> >Response on Innovation
>
> We sincerely appreciate the additional references you provided. These have been cited in our revised manuscript. Below, we elaborate on the innovations of our method compared to previous works.
>
> First, we introduce prior methods and highlight their differences from ours:\
> (1) We start with translation-based approaches. The motivation of [1, 2] is to **present ASR transcripts of input speech while performing translation. To achieve synchronized generation of translations and transcripts, these methods adopt interleaved output of ASR and translation texts at different granularities. However, they do not explore using the model's multi-output capability to implement core policy-decisions in simultaneous translation—such as selecting appropriate generation timings**. They also **lack truncation optimization for long-duration speech inputs, inevitably leading to degraded translation performance with extended inputs [3]**. Additionally, the attention mechanism of these models results in high computational costs when handling long input or output sequences. In terms of results, **these methods fail to demonstrate the translation generation advantages of using ASR as an intermediate step. Instead, better translation performance is achieved by first conducting speech translation followed by ASR** [1, 2]. Thus, **the methods in [1, 2] differ significantly from ours not only in approach but also in conclusions, which contradict existing LLMs-based works [4]**.\
> (2) The work in [5] attempts to realize synchronized generation of ASR transcripts and translation words through word-by-word alternating generation between the source and target languages. However, **its primary focus remains on the interleaved strategy between source and target**. Even after generating source and target words, it requires reordering all target words to ensure the readability of the target translation. Consequently, **this scheme struggles to provide appropriate translations in real-time scenarios and does not address generation or truncation policies in simultaneous translation, making it difficult to extend to long-duration speech inputs**. Therefore, this method differs greatly from ours in motivation, approach, and results.\
> (3) **Next, we discuss the approaches of speech LLMs**. Methods such as [6, 7, 8] are end-to-end speech LLMs. During training, they use interleaved speech and text tokens within a single sample to **maximize the transfer of textual capabilities to speech**. In generation, **they leverage text-guided speech generation to achieve better speech-output performance**. However, **their core focus is on improving the quality of speech responses, not on using multi-stage outputs to enable the model to acquire generation policies for autonomous decision-making on generation timings**. Furthermore, these models lack strong translation capabilities, making them unsuitable for high-performance simultaneous translation. **They also fail to handle long-duration speech inputs effectively due to the absence of truncation strategies. Notably, the paper [8] you mentioned was published after our submission.**
>
> The motivation of our method is to efficiently equip speech large models with long-duration simultaneous translation capabilities. Using Speech Chain-of-Thought (CoT) as the carrier, **we leverage its multi-stage outputs to implement the generation policy, truncation policy, and translation generation required for simultaneous translation**. **The generation policy** ensures the model can detect speech inputs in real-time, generating translations immediately after valid speech is received to balance low latency and high translation quality. **The truncation policy** promptly discards unnecessary historical speech inputs and translations, reducing computational costs and processing delays. **Speech CoT** further enhances the model’s translation generation ability, improving overall translation quality. **The effectiveness of each module has been proved using experiments in our previous Response (3/4)**.
>
> [1] Weller et al. Streaming Models for Joint Speech Recognition and Translation. EACL 2021.\
> [2] Papi, et al. Token-Level Serialized Output Training for Joint Streaming ASR and ST Leveraging Textual Alignments. IEEE ASRU Workshop 2023.\
> [3] Li, et al. Enhancing Document-level Translation of Large Language Model via Translation Mixed-instructions. arXiv:2401.08088.\
> [4] Huang, et al. Speech Translation with Large Language Models: An Industrial Practice. arXiv:2312.13585.\
> [5] Omachi, et al. Align, Write, Re-Order: Explainable End-to-End Speech Translation via Operation Sequence Generation. ICASSP 2022.\
> [6] Nguyen, et al. SpiRit-LM: Interleaved Spoken and Written Language Model. TACL 2025.\
> [7] GLM-4-Voice: Towards Intelligent and Human-Like End-to-End Spoken Chatbot. ICLR 2025.\
> [8] Wu, et al. Chronological Thinking in Full-Duplex Spoken Dialogue Language Models. arXiv:2510.05150.

---

### Official Review · Reviewer_ZGw5 · 2025-11-02

**Soundness:** 4
**Presentation:** 3
**Contribution:** 4
**Rating:** 6
**Confidence:** 4

**Summary:**

This paper introduces StreamUni, a method for streaming speech translation that leverages a large speech language model to achieve various steps (segmentation, policy decision, translation).
The method shows better latency-quality tradeoffs than the baseline and various ablations are presented.

**Strengths:**

* the proposed method is interesting
* the empirical results are strong
* the method is applicable to both streaming and simultaneous ST

**Weaknesses:**

* situate the work better, for example https://arxiv.org/abs/2502.03382 is not cited/compared to
* given that simulst is quite a practical application, add considerations on real-time deployment and clarify whether the evaluation takes computational latency into account

Due to these weaknesses, the current overall rating is a more conservative 6 but the reviewer is looking forward to authors' answers.

**Questions:**

* 066-067: “but also present difficulties in efficiently transferring to newly advanced LSLMs.” Can you elaborate on this particular point?

* 309-311 are you comparing to the competition winners of the annual IWSLT competition? This would ensure that the proposed method is SoTA as claimed in the paper.

* 344: “using Stream SacreBLEU”: what does Stream SacreBLEU refer to?

* Could you elaborate on the anticorrelation between comet and bleu in Table 1?

Typos/Presentation

* Suggest defining more formally the tasks of simulst and streamst. In the current version, the definitions are somewhat implicit (simulst operates on a sentence while streamst operates on longer input (but unclear how long, infinite? etc.))

259: “for a instance”: for an instance

The conclusion is very short (probably to fit the limit), consider expanding.

---

> ### Author Response · Authors · 2025-11-23
> **Response (1/2)**
>
> **Thank you very much for your helpful suggestions.**
>
>
> >Cite the paper [1] and situate our work better?
>
> Thank you for your valuable suggestion. We have added the citation of [1] to both the Introduction and Related Work sections, and have further clarified the positioning of our work in the context of this prior study.
>
> [1] Labiausse, et al. High-Fidelity Simultaneous Speech-To-Speech Translation.
>
> >Consider real-time deployment of your method and clarify whether the evaluation takes computational latency into account?
>
> We sincerely appreciate your insightful feedback. To validate the feasibility of our method for real-world deployment, we explicitly incorporate computational latency into our evaluation framework when conducting experiments on the NVIDIA GeForce RTX 3090 (a consumer-grade GPU). Specifically, we adopt the computation-aware LAAL metric [1, 2], which quantifies the end-to-end latency from the user's speech input to the model's translation output. The table below reports our method's performance on the MuST-C En-De and En-Es datasets, with both computation-aware latency and non-computation-sensitive metrics included for comprehensive comparison.
>
> | MuST-C En-De | | |
> |:--:|:--:|:--:|
> | **Stream LAAL** | 1762.85 | 2223.69 |
> | **Computation-Aware Stream LAAL** | 3193.94 | 3690.02 |
> | **Stream SacreBLEU** | 25.65 | 26.89 |
>
> | MuST-C En-Es | | |
> |:--:|:--:|:--:|
> | **Stream LAAL** | 1802.59 | 2087.39 |
> | **Computation-Aware Stream LAAL** | 3077.07 | 3405.03 |
> | **Stream SacreBLEU** | 29.02 | 30.18 |
>
> **As can be seen from the table above, our method achieves promising performance with a latency of approximately 3 seconds when computational costs are taken into account, and around 2 seconds when computational delays are not considered**. Notably, these results are obtained without leveraging any inference frameworks or advanced GPUs. It is anticipated that the adoption of the aforementioned optimization techniques will enable us to achieve even much lower latency.
>
> [1] Papi,et al. Over-Generation Cannot Be Rewarded: Length-Adaptive Average Lagging for Simultaneous Speech Translation. NAACL 2022 Workshop.\
> [2] Xu et al. CA*: Addressing Evaluation Pitfalls in Computation-Aware Latency for Simultaneous Speech Translation. Findings of NAACL 2025.
>
> >Elaborate on this particular point "but also present difficulties in efficiently transferring to newly advanced LSLMs." in Line 66-67?
>
> Your observation is insightful. The existing methods referenced here [1, 2] typically equip models with real-time generation and policy-decision capabilities by constructing interleaved input-output data. Specifically, they break down traditional offline translation data into fine-grained segments and structure them into the following format:
>
> **"[Input Seg1] [Output Seg1] [Input Seg2] [Blank] [Input Seg3] [Output Seg3]"**\
> Among this format:\
> "**[Input Seg]**" represents the user's input;\
> "**[Output Seg]**" denotes the translated content generated by the model;\
> "**[Blank]**" indicates that no content is generated at the current moment.
>
> During the training phase, such data is used for model training to help the model learn real-time translation generation and policy-decision abilities. However, for the LSLMs to acquire these capabilities, it requires a massive amount of training data and the process of constructing this data consumes substantial resources. Furthermore, each time a new advanced LSLM emerges, it must be retrained using such tailored data. This makes it expensive to develop simultaneous translation models based on these newly advanced LSLMs.
>
> [1] Wang, et al. Conversational SimulMT: Efficient Simultaneous Translation with Large Language Models. IWSLT 2025.\
> [2] Cheng, et al. Towards Achieving Human Parity on End-to-end Simultaneous Speech Translation via LLM Agent.

---

> ### Author Response · Authors · 2025-11-23
> **Response (2/2)**
>
> >Compare to the competition winners of the annual IWSLT competition? This would ensure that the proposed method is SoTA as claimed in the paper.
>
> Your comment is insightful. Our method has not been compared with the approaches from the IWSLT competition, primarily for two reasons.
>
> On one hand, the training dataset adopted by the methods in the IWSLT competition [1] is significantly larger than the test dataset used in our method, which would lead to potential unfairness in a direct comparison of the final test results.
>
> On the other hand, there are inconsistencies between the test datasets and the segmenter modules employed in the testing processes [2], making direct comparison infeasible. Instead, our method follows the experimental settings of previous works and has been compared with approaches such as StreamAttP, StreamAttFW [3], and Phi-4-VAD to validate the effectiveness of our training scheme, truncation policy, and translation capabilities. We have revised the constraints for claiming the state-of-the-art (SoTA) performance of our method in the paper accordingly, emphasizing that this claim holds under consistent settings including identical training data and test conditions.
>
> [1] https://iwslt.org/2025/offline#training-data-and-data-conditions \
> [2] Tan, et al. NAIST Simultaneous Speech Translation System for IWSLT 2025. IWSLT 2025.\
> [3] Papi, et al. StreamAtt: Direct Streaming Speech-to-Text Translation with Attention-based Audio History Selection. ACL 2024.
>
> >What does Stream SacreBLEU in 344 refer to?
>
> In the setup of our method, SacreBLEU is used to evaluate the SimulST task, i.e., sentence-level translations. Correspondingly, **Stream SacreBLEU is employed for the StreamST task, which measures the quality of document-level translations**. To compute Stream SacreBLEU, we first align document-level translations with sentence-level references using mWERSegmenter, converting the document-level outputs into sentence-level translations. We then calculate the SacreBLEU score between these sentence-level translations and the references, leading to the proposed Stream SacreBLEU. Detailed elaboration on this method is provided in the revised manuscript.
>
> >Elaborate on the anticorrelation between comet and bleu in Table 1?
>
> Table 1 shows that the truncation policy of our method outperforms that of MuST-C on the COMET metric but is slightly inferior on SacreBLEU. To investigate this discrepancy, we analyze several cases and find that while our method sometimes produces translations inconsistent with the ground-truth in wording, it maintains semantic consistency. **Such outputs are regarded as negative examples by BLEU but positive ones by COMET—explaining why our method achieves lower BLEU scores yet higher COMET performance despite minor deviations from the ground-truth in expression**.
>
> Additionally, we observe that translations separated by the truncation policy in our method tend to be longer. We therefore calculate the average speech duration and average translation length segmented by different truncation policies.
>
> | MuST-C En-De | | |
> |:--:|:--:|:--:|
> | **Settings** | **Avg Duration** | **Avg Translation Length** |
> | StreamUni | 9.34 | 27.91 |
> | MuST-C Official Truncation | 5.77 | 15.60 |
>
> Our findings indicate that our method can leverage longer-range information to facilitate translation. We hypothesize that it exhibits greater consistency in translating nouns: even if a preceding part contains formal errors but conveys the correct meaning, the subsequent translation retains this inconsistent word. This may be another reason for its slightly lower BLEU performance compared to MuST-C’s truncation policy.
>
> >Supplement the definitions of "SimulST" and "StreamST", and expand the conclusion section accordingly?
>
> Following your suggestions, we have revised the original paper.
>
> **Regarding the above responses we provided to you, we have already added and revised the relevant content in the paper. If our responses meet your satisfaction, we would greatly appreciate it if you could consider raising the score.**

---

### Official Review · Reviewer_merF · 2025-11-03

**Soundness:** 2
**Presentation:** 2
**Contribution:** 2
**Rating:** 4
**Confidence:** 5

**Summary:**

This paper presents StreamUni, a streaming translation model designed to handle unbounded speech input. The model operates within a single speech language model that jointly performs transcription, truncation, and streaming translation. Specifically, the system first transcribes incoming speech, then performs truncation based on detected silences and sentence boundaries, and finally generates translations in a streaming fashion using a wait-k like policy conditioned on partial speech, prior transcripts, and partial translations. Experiments on the MuST-C dataset demonstrate that StreamUni achieves a superior quality–latency trade-off compared to baseline systems when processing unbounded speech streams. Additional ablation studies highlight the effectiveness of the training data mixture and the impact of truncation strategy on performance.

**Strengths:**

1. **Unified Speech Language Model for Multiple Tasks:**

    StreamUni effectively fine-tunes a single speech language model to perform three key actions—transcription, truncation, and streaming generation—within a unified framework. This design demonstrates strong empirical performance despite being trained on relatively limited data.

2. **Reinforcement of Data Mixing Importance:**

    Although the benefit of mixing offline and streaming data during training for streaming translation has been discussed in prior work [1], this paper provides additional empirical evidence supporting its effectiveness, further validating this important training strategy.


[1] Fu, Biao, et al. "Efficient and Adaptive Simultaneous Speech Translation with Fully Unidirectional Architecture." *arXiv preprint arXiv:2504.11809* (2025).

**Weaknesses:**

1. **Flaws in Experimental Design:**
    - **Unfair baseline comparison:** The baselines (e.g., EDAtt for SimulST and StreamAtt for StreamST) rely on offline speech translation models trained with significantly weaker base architectures than Phi-4-Multimodal, which serves as the backbone of StreamUni. Since even their *offline translation quality* differs substantially, the comparison does not fairly isolate the advantage of the proposed method itself.
    - **Lack of cascade baseline:** The entire StreamUni pipeline—comprising transcription, truncation, and wait-k–style streaming generation—could be replicated with a cascaded ASR + MT architecture. Including such a comparison would provide a stronger and more interpretable baseline.
2. **Inaccuracies and Ambiguities in Writing:**
    - In the abstract, the authors claim that prior works translate using only limited contextual information, yet StreamUni also conditions translation on a restricted context window, potentially leading to term inconsistency across long-form speech.
    - **Lines 68–70:** The claim that prior methods rely on an upstream segmentation model is inaccurate, as [2] demonstrates direct streaming translation on unbounded speech using attention-sink techniques, without requiring segmentation.
    - **Equations (1) and (2):** These equations should use the product of probabilities, not their summation.
    - **Lines 175–179:** The explanation of the truncation process is ambiguous and only becomes clear after reading the following “Truncation Policy” subsection; this section should be reorganized for clarity.
    - **Figure 1:** The diagram provides little information into what the generation policy entails or how it operates.
    - **Figure 4:** The term “Partial Streaming” is undefined—its precise meaning should be clarified.
    - **Table 1:** The label “Human segmentation” is misleading, since MuST-C segmentations are automatically derived, not human-annotated.

[2] Siqi Ouyang, Xi Xu, and Lei Li. 2025. InfiniSST: Simultaneous Translation of Unbounded Speech with Large Language Model. In *Findings of the Association for Computational Linguistics: ACL 2025*, pages 3032–3046, Vienna, Austria. Association for Computational Linguistics.

**Questions:**

Do the authors have any intuition or analysis on why the model’s truncation strategy outperforms MuST-C’s segmentation in terms of COMET scores?

---

> ### Author Response · Authors · 2025-11-23
> **Response**
>
> **Thank you very much for your insightful perspectives and helpful suggestions.**
>
> >Unfair baseline comparison: The baselines (e.g., EDAtt for SimulST and StreamAtt for StreamST) rely on offline speech translation models trained with weaker base architectures than Phi-4-Multimodal? The comparison does not fairly isolate the advantage of the proposed method itself?
>
> (1)Your suggestions are of significant value. For the SimulST task, we conduct additional comparisons with EASiST [1], which employs Llama-3-8B-Instruct as its backbone. Experimental results demonstrate that our method achieves a performance improvement of over 1 BLEU on both MuST-C En-De and MuST-C En-Es tasks under the same latency constraints for the SimulST task. **This validates the effectiveness of the generation policy in our approach**.
>
> | MuST-C En-De | | | | |
> |:--:|:--:|:--:|:--:|:--:|
> | **StreamUni** | LAAL | 1683.09 | 1808.49 | 2350.03 |
> | | SacreBLEU | 26.98 | 28.00 | 30.55 |
> | **EASIST** | LAAL | 1620 | 2310 | 3230 |
> | | SacreBLEU | 25.36 | 27.28 | 28.61 |
>
> | MuST-C En-Es | | | | |
> |:--:|:--:|:--:|:--:|:--:|
> | **StreamUni** | LAAL | 1702.22 | 1960.39 | 2463.44 |
> | | SacreBLEU | 28.92 | 31.56 | 33.39 |
> | **EASIST** | LAAL | 1910 | 2430 | 3460 |
> | | SacreBLEU | 30.12 | 31.00 | 31.92 |
>
>
> (2) For the StreamST task, we compare the truncation policy mechanism of StreamUni with the method using silero-vad, based on the same speech LLMs. The experimental results are shown in the following table. By incorporating semantic information, our method avoids the model simply truncating historical content based on whether a person is speaking. This not only prevents inappropriate decision-making timings but also yields better translation quality. **This validates the effectiveness of truncation policy of our method**.
>
> | MuST-C En-De | | | |
> |:--:|:--:|:--:|:--:|
> | **StreamUni** | Stream LAAL | 1762.85 | 2223.69 |
> | | Stream SacreBLEU | 25.65 | 26.87 |
> | **Phi-4-VAD** | Stream LAAL | 1774.12 | 2204.71 |
> | | Stream SacreBLEU | 24.44 | 24.37 |
>
> [1] Fu, Biao, et al. "Efficient and Adaptive Simultaneous Speech Translation with Fully Unidirectional Architecture." arXiv preprint arXiv:2504.11809 (2025).
>
> >Include a cascaded baseline?
>
> Thank you for your valuable suggestions. To verify the effectiveness of our method, we additionally conduct a cascaded experiment: we **adopt Whisper-Large-V3 as the ASR model and fed its outputs into Phi-4-Mini-Instruct for translation**. The prompt used for Phi-4-Mini-Instruct is: **"Translate the English text to German text based on the given German translation. English text: {cot_asr}. \n\n German translation: {cot_st}"**. We compare this cascaded scheme with our method, and the results are shown in the table below.
>
> | MuST-C En-Es | | | |
> |:--:|:--:|:--:|:--:|
> | **StreamUni** | Stream LAAL | 2087.39 | 2256.91 |
> | | Stream SacreBLEU | 30.17 | 30.67 |
> | **Cascaded** | Stream LAAL | 2384.68 | 2559.65 |
> | | Stream SacreBLEU | 18.69 | 21.93 |
>
> As indicated in the table, our model outperforms the cascaded model significantly, which we attribute to two factors. First, Phi-4-Mini-Instruct sometimes fails to fully follow instructions—on occasion, it retranslates instead of continuing based on the provided translation, and at other times it merely restates the input rather than performing translation. Second, our StreamUni benefits from the effectiveness of its policy-decision and training scheme.
>
> >Inaccuracies and Ambiguities in Writing?
>
> We greatly appreciate your pointing out the ambiguities and omissions in the writing. Regarding the issues you raised, we have supplemented and revised the relevant content in the paper. We have also added citations to the papers [1, 2] you mentioned.
>
> [1] Fu, Biao, et al. Efficient and Adaptive Simultaneous Speech Translation with Fully Unidirectional Architecture.\
> [2] Siqi Ouyang, et al. InfiniSST: Simultaneous Translation of Unbounded Speech with Large Language Model. Findings of ACL 2025.
>
> > Why the model's truncation strategy outperforms MuST-C's segmentation in terms of COMET scores?
>
> To explore the underlying reasons, we conduct a detailed analysis of the average duration of truncated speech chips and average translation length obtained from two truncation policies. The results are shown in the table below.
>
> | MuST-C En-De | | |
> |:--:|:--:|:--:|
> | **Settings** | **Avg Duration** | **Avg Translation Length** |
> | StreamUni | 9.34 | 27.91 |
> | MuST-C | 5.77 | 15.60 |
>
> We find that the truncation policy of our method enables the model to process longer speech segments and reference lengthier historically generated translations. This provides sufficient context for the model during translation, thus outperforming the truncation method of MuST-C.
>
> **Regarding the above responses we provided to you, we have already added and revised the relevant content in the paper. If our responses meet your satisfaction, we would greatly appreciate it if you could consider raising the score.**

---

> > ### Comment · Reviewer_merF · 2025-11-25
> >
> > Thank you for the response and for adding the additional experiments.
> >
> > Including EASiST helps make the evaluation more complete; however, Phi-4 and Llama are still fundamentally different models. This does not directly address my concern about the fairness of the comparison.
> >
> > Additionally, could you provide some concrete examples to illustrate the claim: “By incorporating semantic information, our method avoids the model simply truncating historical content based on whether a person is speaking”? I would like to better understand how this manifests in practice.

---

> > > ### Author Response · Authors · 2025-11-27
> > > **Further Response to Reviewer merF**
> > >
> > > >More experiments to demonstrate the effectiveness of the generation policy?
> > >
> > > To demonstrate the effectiveness of our generation policy, we first compare it with EASiST (which adopts Llama-3-8B as the base LLM) in our preliminary experiments. **For further validation, we implement an alternative generation policy—dubbed Phi-4-Waitk—on top of our simultaneous translation model**. Unlike policies that rely on spoken information in speech, Phi-4-Waitk determines the number of target words to generate by directly counting the number of input speech chunks. We evaluate the SimulST performance of both models on the MuST-C En-De and MuST-C En-Zh datasets to verify the efficacy of our generation policy.
> > >
> > > | MuST-C En-De SimulST Task | | |
> > > |:--:|:--:|:--:|
> > > | **StreamUni** | LAAL | 2014.90 |
> > > | | SacreBLEU | 29.29 |
> > > | **Phi-4-Waitk** | LAAL | 2460.40 |
> > > | | SacreBLEU | 27.53 |
> > >
> > > | MuST-C En-De StreamST Task | | | |
> > > |:--:|:--:|:--:|:--:|
> > > | **StreamUni** | LAAL | 1633.86 | 2038.431 |
> > > | | SacreBLEU | 34.73 | 36.83 |
> > > | **Phi-4-Waitk** | LAAL | 1762.892 | 2250.476 |
> > > | | SacreBLEU | 28.62 | 32.05 |
> > >
> > > As shown in the tables above, our method achieves superior generation quality with lower latency across both tasks using the same speech LLMs, thereby validating the effectiveness of our proposed generation policy.
> > >
> > >
> > > >Provide some concrete examples to illustrate the claim: "By incorporating semantic information, our method avoids the model simply truncating historical content based on whether a person is speaking"?
> > >
> > > We present a case study from the 3rd sample in the sentence-level test set of MuST-C En-De to support this claim.
> > >
> > > 1. For this example, the truncation method in the MuST-C En-De dataset is provided with the following Ground-Truth annotations:\
> > > **Source sentence:** "Now in our town, where the volunteers supplement a highly skilled career staff, you have to get to the fire scene pretty early to get in on any action."\
> > > **Target Sentence:** "Nun, in unserer Stadt, in der Freiwillige eine hochqualifizierte Berufsfeuerwehr unterstützen, muss man ziemlich früh an der Brandstelle sein, um mitmischen zu können."
> > >
> > > 2. VAD-Based Truncation Scheme\
> > > This scheme splits the source sentence into three segments based on speaker pauses, leading to segmented translations:\
> > > **Source sentence:**\
> > > (1)"Now in our town, where the volunteers supplement a highly skilled career staff."\
> > > (2)"You have to."\
> > > (3)"Get to the fire scene pretty early to get in on any action."\
> > > **Translation:**\
> > > (1)"In unserer Stadt ja, da helfen ehrenamtliche Leute mit, neben den professionellen Mitarbeitern."\
> > > (2)"ich muss mich umstellen."\
> > > (3)"Musst ziemlich früh am Brandort da sein, wenn du mitmachen willst."
> > >
> > > The VAD-based approach truncates the source sentence arbitrarily at pauses, disrupting semantic coherence. For example, Segment (2) ("You have to.") is semantically incomplete, resulting in an irrelevant translation ("Ich muss mich umstellen" = "I have to adapt").
> > >
> > >
> > > 3. Our Semantic-Aware Truncation Policy (StreamUni)\
> > > Our method preserves semantic integrity by aligning truncation policy with the Ground-Truth truncation, yielding a coherent and accurate translation:\
> > > **Source Sentence**: "Now in our town, where the volunteers supplement a highly skilled career staff."\
> > > **Translation**: "In unserer Stadt, wo die Freiwilligen das hochqualifizierte Berufspersonal ergänzen, muss man ziemlich früh am Brandort eintreffen, um an der Aktion teilzunehmen."
> > >
> > > This case study demonstrates the effectiveness of StreamUni's semantic-aware truncation policy in mitigating the limitations of VAD-based approaches.

---

> > > > ### Comment · Reviewer_merF · 2025-11-27
> > > >
> > > > Why Phi-4-WaitK differs with StreamUni even in SimulST setting? StreamUni also adopts the wait-k like strategy and I suppose truncation does not happen frequently in this setting.

---

> > > > > ### Author Response · Authors · 2025-11-27
> > > > > **Further Response to Reviewer merF**
> > > > >
> > > > > In SimulST, the Phi-4-Wait-k policy dictates that the first target word is only generated and output after receiving k speech chunks from the source. Subsequently, it follows a "read one speech chunk, generate one target word" paradigm. However, our StreamUni framework differs fundamentally from this approach. Specifically, our method leverages Speech Chain-of-Thought (CoT) to obtain the speaker's ASR transcript and initiates target output only after accumulating k source words. Thereafter, it adheres to a "detect one source word, output one target word" mechanism. Consequently, compared to Phi-4-Wait-k, StreamUni dynamically determines the optimal timing for target word generation, achieving a superior trade-off between translation latency and quality.

---

> > > > > > ### Comment · Reviewer_merF · 2025-11-27
> > > > > >
> > > > > > I see. This is more of less like ASR then do wait-k SimulMT. Then I'm a bit curious why cascade baseline you run before with whisper + Phi-4 is so bad? How do you implement it? A standard way could be like let ASR produce multiple candidates, translate each of them, apply local agreement policy.

---

> > > > > > > ### Author Response · Authors · 2025-11-28
> > > > > > > **Further Forth Response to Reviewer merF**
> > > > > > >
> > > > > > > Yes, we conduct the experiment on the Whisper + Phi-4-Mini-Instruct pipeline in accordance with the scheme you mentioned. To investigate the question of why this cascaded approach yields relatively poor performance compared to StreamUni, **we evaluate the performance of cascaded method in a StreamST scenario where its generation policy is disabled, with translations are only generated at the truncation timing**. This scenario corresponds to the offline performance in traditional SimulST scenarios.
> > > > > > >
> > > > > > > We find that the translation performance of Phi-4-Mini-Instruct under this setting is 26.81. In contrast, our StreamUni achieve a performance of 31.59 in the same scenario. Therefore, we conclude that **the first reason why the cascaded scheme performs worse than our StreamUni is that Phi-4-Mini-Instruct exhibits weak English-to-German (En-De) translation capabilities when not subjected to meticulous fine-tuning**.
> > > > > > >
> > > > > > > Another contributing factor is that we observe Phi-4-Mini-Instruct often fails to adhere to user instructions effectively when using the following prompt.\
> > > > > > > **"Continue to translate the English text to German text based on the given German translation. English text: {cot_asr}. \n\n German translation: {cot_st}"**
> > > > > > >
> > > > > > > **Instead of continuing the translation based on the already generated translations, it sometimes attempts to restart the translation process. This issue further degrades the performance of the cascaded scheme in simultaneous interpretation scenarios.**

---

### Note · Authors · 2026-01-05

I have read and agree with the venue's withdrawal policy on behalf of myself and my co-authors.